# Examination of the Effects of Domestic Water Buffalo (*Bubalus bubalis*) Grazing on Wetland and Dry Grassland Habitats

**DOI:** 10.3390/plants12112184

**Published:** 2023-05-31

**Authors:** Attila Fűrész, Károly Penksza, László Sipos, Ildikó Turcsányi-Járdi, Szilárd Szentes, Gabriella Fintha, Péter Penksza, Levente Viszló, Ferenc Szalai, Zsombor Wagenhoffer

**Affiliations:** 1Institute of Agronomy, Hungarian University of Agriculture and Life Science, Páter Károly u., 2100 Gödöllő, Hungary; furesz.attila.zoltan@phd.uni-mate.hu (A.F.); penksza.karoly@uni-mate.hu (K.P.); ildikojardi@gmail.com (I.T.-J.); 2Department of Postharvest, Commercial and Sensory Science, Institute of Food Science and Technology, Hungarian University of Agriculture and Life Sciences, Villányi út, 1118 Budapest, Hungary; penksza.peter@nak.hu; 3Institute of Economics, Centre of Economic and Regional Studies, Tóth Kálmán u., 1097 Budapest, Hungary; 4Animal Breeding, Nutrition and Laboratory Animal Science Department, University of Veterinary Medicine Budapest, István u., 1078 Budapest, Hungaryzsombor.wagenhoffer@univet.hu (Z.W.); 5Doctoral School of Biological Sciences, Hungarian University of Agriculture and Life Science, Páter Károly u., 2100 Gödöllő, Hungary; fintha.gabriella@phd.uni-mate.hu; 6MTA-EKE Lendület Environmental Microbiome Research Group, Eszterházy Károly University, Leányka u., 3300 Eger, Hungary; 7Pro Vértes Nature Conservation Foundation, Kenderesi út, 8083 Csákvár, Hungary; 8The Water Buffalo Reserve of Mátra, Lapos Tanya, 3064 Pásztó, Hungary

**Keywords:** control of woody species, *Festuca* sp., grassland management value, invasive species, pasture, turf management value

## Abstract

In nature conservation today, there is a global problem with the aggressive expansion of invasive plant species and the conservation of valuable grassland vegetation. Based on this, the following question has been formed: Is the domestic water buffalo (*Bubalus bubalis*) appropriate for managing various habitat types? How does grazing by water buffalo (*Bubalus bubalis*) affect on grassland vegetation? This study was carried out in four areas of Hungary. One of the sample areas was in the Mátra Mountains, on dry grassland areas where grazing had been applied for two, four and six years. The other sample areas were in the Zámolyi Basin, where wet fens with a high risk of *Solidago gigantea* and in a typic Pannonian dry grassland were investigated. In all areas, grazing was carried out with domestic water buffalo (*Bubalus bubalis*). During the study, we carried out a coenological survey, examining the change of cover of plant species, their feed values and the biomass of the grassland. According to the results, both the number and cover of economically important grasses (from 28% to 34.6%) and legumes (from 3.4% to 25.4%) increased in Mátra as well as the high proportion of shrubs (from 41.8% to 4.4%) shifted toward grassland species. In the areas of the Zámolyi Basin, invasive *Solidago* has been suppressed completely, the pasture has been converted completely (from 16% to 1%) and the dominant species has become *Sesleria uliginosa*. Thus, we have found that grazing with buffalo is suitable as a habitat management method in both dry grasslands and wet grasslands. Therefore, in addition to its effectiveness in the control of *Solidago gigantea*, grazing with buffalo is successful in both nature conservation and economic aspects of grassland vegetation.

## 1. Introduction

Grasslands are habitats characterized by native grasses and dicotyledonous herbs and with a low ratio of woody species. This area type occurs all over the world. Semi-natural grasslands are grasslands modified by human activities. Nature conservation has to fight against losing valuable grasslands. Grazing is practiced in areas with a poor quality because many of which have been left fallow and usually turned into weedy areas [1]. Several grasslands are degraded, weedy and shrubbed without conservation management, human interventions [2,3,4,5] or grassland management practices [6]. Furthermore, grassland restoration and conservation have recently been gaining increased importance [3,7,8,9,10] as one of the most applied habitat restoration interventions today. Besides being advisable after re-grazing, grazing can be applied individually for grassland management after the establishment of the grassland structure [6,11,12,13,14]. 

Most of the grazing is done by cattle in the management of nature reserves in order to preserve the biodiversity of grasslands and to restore grasslands in many kinds of habitats [4,15,16,17,18]. Grazing by cattle is suitable for native biodiversity conservation in grasslands in general because it does not eat so selectively resulting in semi-natural habitats, unlike grazing with horses, sheep or goats [19,20,21]. In many cases, native biodiversity and their valuable habitats are threatened by invasive species.

Invasive non-native species have a large negative impact on not only natural ecosystems but also on the economy and human health [22,23,24]. Invasive plant species can cause a decline in biodiversity [25,26,27] because they can change biodiversity, environmental conditions and community structures through degeneration of plant communities, landscape changes and ecosystem impacts [24]. These aggressive plants are frequently monodominant and expand quickly, reducing the biodiversity of agricultural fields and natural vegetation [28,29,30]. 

One of the most aggressive invasive plants is the giant goldenrod (*Solidago gigantea*) [31] which is derived from North America [32]. *Solidago gigantea* becomes dominant easily, suppressing other valuable plant species and it is found growing, expanding aggressively in ruderal habitats, left fallow, meadows, pastures, fields, forests, on roadsides, riversides, trenches, etc. [23,30,33,34]. In addition, this species influences the PH of the soil negatively, which can also lead to a reduced soil biodiversity [30,35], although according to other investigations, plants do not affect the pH of the soil significantly [36]. Thus, it is necessary to introduce effective management, which can both suppress alien species and restore community functions [37]. One of the main nature conservation management is mowing [38,39,40,41], but some studies have been carried out on grazing too. Moreover, a combination of management can increase management efficiency [42]. Nevertheless, rhizomes of plant species can spread and regenerate rapidly [29,43,44], so the people who carry out the management should avoid the fragmentation and dispersal of rhizomes. In addition, based on the results of Nagy et al. [45], grazing by sheep has a less positive effect on native plant community survival than grazing by cattle, so cattle grazing is appropriate in general. Furthermore, combined management of grazing by cattle and single mowing is not effective because mowing decreases the efficacy of grazing [45], although numerous other studies revealed that long-term mowing may be an adequate option for *Solidago gigantea* removal [38,39,40]. According to the research of Hajnáczki et al. [8], goats also eat giant goldenrod (*Solidago gigantea*), but only in limited amounts because the high saponin content of the plant [46] reduces some digestible nutritional values [47].

Nowadays, the importance of grazing by domestic water buffalo (*Bubalus bubalis*) has been growing for habitat management and economic reasons [48,49,50,51].

Several researchers claimed that grazing by domestic water buffalo (*Bubalus bubalis*) has shown positive effects on grassland management and nature conservation. It is recognized globally that water buffaloes (*Bubalus bubalis*) are appropriate for grazing in wetlands in order to conserve biodiversity [52,53].

However, their significance may not only be outstanding in wetland habitats, but they can be important in dry habitats as well. Studies have been made in Australia, where water buffalo (*Bubalus bubalis*) have been grazed on the savanna [54]. The reason why water buffaloes (*Bubalus bubalis*) might be able to graze in also dry habitats is possible as they have better digestibility than cattle [55,56] because they are more efficient in the use and transformation of feed than other draught animals [57,58]. The main advantageous characteristic of the water buffalo (*Bubalus bubalis*) is its special ability to subsist on coarse feed, straw and crop residues and to transform these materials into protein-rich lean meat that is low in cholesterol [59]. On the other hand, they eat less selectively in order to fill their stomachs among grazing animals [60]. Nevertheless, data on water buffaloes’ dietary behavior is very limited in terms of professional literature [61,62]. For example, the use of trees as a source of feed for ruminants is limited to understanding of their positive impact on production systems [63]. Galloso-Hernández et al. [64] conducted research on the feeding of buffaloes on woody plants, but they only fed the leaves of trees to the buffaloes.

A hypothesis based on the reviewed literature is that as buffalo have a better digestive capacity than cattle, they may be able to graze on giant goldenrods with a high saponin content and woody plants such as shrubs. It would be necessary because *Solidago gigantea* suppresses plants that are useful for turf management too.

Based on these, in our research, we sought answers to the following questions: (1)Whether grazing by water buffalo (*Bubalus bubalis*) can be applicable for the control of shrub encroachment?(2)How does grazing by water buffalo (*Bubalus bubalis*) affect the species composition and structure of the grasslands in a typical Pannonian dry grassland, a typical wet fen and a grassland in a mid-mountain area, mainly agronomically (turf management)?(3)Is grazing by water buffalo (*Bubalus bubalis*) effective for the suppression of the invasive *Solidago gigantea*?

## 2. Results

### 2.1. Species Composition and Structure of Vegetation

During the assessment of the data, six functional groups were distinguished according to functional groups.

Based on pair-wise comparisons, the cover values were found to be significant (Kruskal–Wallis test) and comparisons by Dunn’s post hoc test with Bonferoni correction were used to determine non-different and different cover values by units of groups, sample area and year.

Based on the histogram of the mean cover of the useful grasses, we can see that useful grass species appeared in the control area over time between only 10–12.5% (Figure 1). However, it can be seen that the proportion of the grass species in the pasture in Csákvár changed significantly with the passage of time. At the start of treatment, the mean cover of useful grasses was only 18.7%, but at a later period, the mean cover of the area was already much higher. For example, one year later the mean cover increased to 66%, but already in 2021, it was 79.5%. Among the useful grasses, *Sesleria uliginosa* showed one of the most significant changes, ranging from 0 to 25% cover (Appendix A). Similarly, it happened with the proportion of useful grasses in ridge planting areas which at the beginning of treatment also was quite low (20.7%), but with constant cover growth after seven years, it was 57.5%. In the mid-mountain area, there was also a slight rise over time as the ratio of useful grass species of the sample area which was grazed for two years was 28%, but where it was grazed for six years, it was 34.6%. Results of statistical analysis showed that the properties of the control area differ most significantly from the data from 2015, 2017, 2019 and 2021 of the pasture in Csákvár. Moreover, the proportion of the data from 2021 of the area where ridge planting had been used differed particularly from the control area, but it was less significant than the pasture data. Based on the results of the recorded data of the pastures in 2021, it would need about seven years for the fen pasture to differ with full significance in terms of useful grasses. In addition, in the case of the ridge area, seven years were not yet sufficient to significantly differ from the control area in terms of useful grasses. Likewise, in the case of the recorded data of the Szurdokpüspöki, six years were not enough to show a significant difference from the control area. 

Based on the histogram of the mean cover of the non-useful grasses, there are not many significant differences between the values of the areas (Figure 2). The values of the control area did not differ significantly from the values of none of the areas, except for the records of the control area (2015–2021) and the record of the pasture in Csákvár from 2017. Nevertheless, the records of the pasture in Csákvár and the records of the area farmed with ridge planting in 2015–2019 differed significantly from the records of the area which had been managed for four years in the Szurdokpüspöki.

Based on the histogram of the mean cover of the legume species, values of the control area differed significantly only from the area managed for six years in the Szurdokpüspöki (Figure 3). Data from the pasture in Csákvár from 2014 differed significantly from the recorded data from 2021, also values from the area with ridge planting from 2019 and 2021 and finally from the areas in Szurdokpüspöki which were managed for four and six years. Furthermore, it can be seen that the recorded data of the area with ride planting from 2014 and 2021 were already significantly different. Hence, it can be seen that due to grazing, legume species in the managed sample areas increased over time and that a significant difference emerged in the sample areas of Csákvár after seven years.

Based on the histogram of the mean cover of the invasive species (Figure 4), it can be seen that the untreated control areas showed a homogeneously high cover of giant goldenrod (*Solidago gigantea*). Among the treated areas in Csákvár, significant changes were observed in the fen pasture from 2017 and significant differences were noted in the area with ridge planting from 2019. Thus, management effects against the invasive species may be evident earlier in the fen pasture than in the formerly cultivated area. It can be concluded that grazing of domestic water buffaloes (*Bubalus bubalis*) might be able to suppress the cover of giant goldenrod (*Solidago gigantea*) after approximately 4–6 years.

Based on the histogram of the mean cover of the shrub species, there is no significant difference between the values of the shrub cover in the control area and the values in none of the areas (Figure 5). In contrast, the data from 2014 and 2015 in the pasture in Csákvár and the values from 2014 in the cultivated area were significantly different from the area in Szurdokpüspöki, that had been grazed for two and four years. In the sample areas of the Szurdokpüspöki, it can be seen that the cover of shrubs was decreasing, but there is no sign of a statistically significant difference between their data. However, it could be possible that further monitoring of the effect of the management will lead to significant results from the shrub aspect.

Based on the histogram of the mean cover of the other species (Figure 6), significant differences were observed between the records of the control area in 2014–2017 and 2021, showing that the data of the pasture in Csákvár in 2019 and 2021, and the data of the cultivated area in 2021 differed significantly. On the other hand, considering only the 2014 and 2021 data of the control area, in addition to these records, the former cultivated area also differed significantly from the 2019 record. In addition, the records of the pasture from the initial study period were significantly different from the 2019 and 2021 records. 

### 2.2. Results of Biomass

In the control area, although yields were between 36.4 and 41.7 t/ha in each year of the survey, the high yields of *Phragmites communis* and *Solidago gigantea* resulted in negative values (−0.47 to −0.26) every year (Appendix A), meaning very poor forage quality, according to Balázs feed value. Although these two species provide good quality feed when they are young, they senesce very quickly, and after blooming they produce high fiber content, especially lignin, so that the period of time before they could be used by the animal is very short, as well as their large size, allelopathy and well-developed root system, inhibiting the reproduction of species of higher nutritional value. Based on statistical assessments, there were no significant changes in total cover, total yields or feed value over the studied period (2014–2021).

Due to the closure of the grassland, there was a very dramatic increase in the yield of pasture. Lower yields in 2021 were caused by a dry year. The improvement of forage quality (1.63 → 2.3) (which means good quality) was influenced by a reduction in the number of species and cover of species with poor feed value or harmful species. In addition, *Agrostis stolonifera* and *Alopecurus pratensis* actually produced greater yields than in 2019. As a result of grazing, the production of *Phragmites communis* and *Solidago gigantea* also showed a constant, sharp decrease from the second year of the study, while the production of grass and legume species with a high forage value multiplied. Based on the statistical assessment of the pasture during the study period, total cover increased from 2014 to 2021. Compared to the values of total cover from 2014 (50%), it increased significantly by 2019 (183%). In the case of total yields, compared to the initial data (6.95 t/ha), there was a significant increase by 2017 (26.5 t/ha). In feed value, compared to 2014 values (1.63), there was a significant increase by 2021 (2.30). Hence, in the case of pasture, a significant improvement was observed firstly in total yields, then in total cover and then in feed value.

This pattern was also observed in the area where there was a ridge planting. The quality of feed varied from −0.62 to 2.60, i.e., it almost reached the medium category from the very poor category. The growth of the −2 category was contributed by many species with low cover. Based on the statistical assessment of the ride planting during the study period, total cover did not differ significantly from 2014 (123%) to 2015 (116%) but by 2021 it already did (170%). In the case of total yields, there was no significant difference from 2014 (28.8 t/ha) to 2017 (19.3 t/ha), there was a significant change only by 2021 (21.2 t/ha). In feed value, there was a constant increase (2014–2021), compared to the data of 2014 (−0.62), a significant improvement already observed from 2017 (2.47) (Table 1).

In the Szurdokpüspöki, the regression of *Prunus spinosa*, *Rosa canina* and several other shrub species showed a decrease in negative categories, accompanied by a rise in valuable species such as *Festuca arundinacea* and *Lotus corniculatus*. The average feed value of the grassland was 1.2 in the area of two years and 2.64 in the area of six years, which is already medium quality. Based on the statistical assessments, there were no significant changes, except for the feed values because the initial data of the area managed for two years (1.20) differed significantly from data of the area managed for six years (2.64) (Table 2).

## 3. Discussion

Based on the results, grazing by buffalo on wet fens may lead to the appearance of economically useful grasses and legumes after seven years, while it can be expected that about four years are necessary for the invasive monodominant species *Solidago gigantea* to be completely suppressed. In the untreated control area, where *Solidago gigantea* was dominant, it demonstrated well a really invasive behavior reducing biodiversity [25,26,27]. According to the results of grazing on the former cultivated area where there was ridge planting, in order to grow enough legume plant species, seven years would be sufficient, but more time might be needed to appear a sufficient number of useful grasses. To completely remove the studied invasive species, approximately six years would be required. Therefore, we can see that despite attempts to control this invasive plant through mowing, and grazing by various livestock including sheep, goats and cattle [8,45], in the present study it seems that grazing by water buffalo may prove to be the most effective and fastest method [38,39,40,41]. Based on the results of mid-mountain sample areas, six years were not enough to show a significant result in terms of useful grasses, legumes or neither. In contrast, the results showed that there is a positive change in terms of reduction of shrubs and increase of legume species and grass species as well.

Some research confirmed our findings that buffaloes eat woody plants [62,65], but other studies claimed that herbaceous vegetation is highlighted and reported the absence of woody species in the water buffaloes’ diet [66,67]. Considering the results of the observed effect, we can confirm that domestic water buffaloes have a different diet than cattle and sheep [68,69,70]. Grazing with water buffalo is really significant because selective grazing affects the structure and the dynamics of plant communities [67,71,72] which can be important for economic aspect too. It is well confirmed in the literature that buffalo can consume woody plants [73] and the *Solidago* species, which is rich in saponin [46] because the slow passage of food from the digestive system of buffaloes [58] makes them capable to consume low quality forage that is high in fiber [74,75,76,77]. Hence, based on the current investigation, water buffaloes really do not eat so selectively [60].

Overall, there is a lack of previous research on the subject, so it is difficult to conclude confidently from the results. The results of the present research are not representative of a similar effect of grazing by buffalo in different habitats in Hungary or other countries, as there are relatively few data available in comparison to the extent of pasture areas and the number of site types. Furthermore, the present study only includes only seven years of data, and the long-term effects of grazing are still unknown, so it cannot be claimed confidently that the results of the present study are representative. To understand the complex impacts of grazing, it would be necessary to examine grazing from various perspectives. For example, grazing by livestock can affect invertebrate species assemblages too [78]. The relatively short period of the present investigation has shown some significant results, but it is consistent with the conclusions of other researchers that a long-term period of monitoring is needed to monitor the vegetation change in response to a specific management. To know the effects in detail, further studies are needed [10,41,79].

## 4. Materials and Methods

### 4.1. Data Collection and Surveyed Areas

Our study was conducted in the central part of the Carpathian Basin (Figure 7).

Coenological surveys on the sample areas were carried out in 2 × 2 m quadrats. Works were made according to the method of Braun-Blanquet [80], but coverage was given in percentage. We considered the level of the plants in the percentage assessment, too which might cause the total coverage to be above 100%. In the sample areas of Zámolyi Basin in Csákvár, we made 6 surveys in 2014, 2015, 2017, 2019 and 2021. In addition, we made also 6 surveys in the sample areas of Szurdokpüspöki in 2022 (Figure 8). Szurdokpüspöki belongs to Mátra Mountain and it is 140 to 180 m above sea level. 

Species names were given according to the nomenclature of Király [81].

Accordingly, we studied different phases of sandy grasslands in the following sample areas. 

Csákvár:Control, which is left untreated and has a high proportion of individuals of the invasive species;Pasture is a fen grassland which is grazed by domestic water buffalo (*Bubalus bubalis*) since 2013 (stocking rate is 2 buffalo/ha) to manage the cover of the invasive species in order to restore *Molinia* meadows;Ridge is an area where ridge planting used to be practiced and recently has been restored through grazing by buffalo since 2013 (stocking rate is 1 buffalo/ha) to manage the area in order to restore typic Pannonian dry grassland.

In the sample areas of the Szurdokpüspök in Mátra Mountain, the areas of scrub were treated using shredding. Then, water buffalo were introduced to control the scrub encroachment and stimulate the establishment of semi-natural grassland in order to restore calcareous rocky steppes (stocking rate is 0.5 buffalo/ha).

I. sample area was grazed by water buffalo for 2 years;II. sample area was grazed by water buffalo for 4 years;III. sample area was grazed by water buffalo for 6 years.The period of grazing was from the 24 April to the 5 November in the sample areas.

### 4.2. Analysis of Biomassa

Biomass investigations were applied [82,83,84,85] in June and in September annually. Parallel with coenological recordings, a 2 × 2 m grassland part was trimmed by a trimmer leaving 7 centimeters high stubble to model the grazing effect of water buffalo.

The grassland production was estimated by Balázs method [82,83] using the following formula:$$P=(\left(M\;-\;s\right)\;\times \;\mathrm{BM}\;\times \;b)\;/\;100.$$

P: yield [Kg/ha]M: grass height [cm]s: stubble height [cm]BM: grass 400 [kg/ha]; alfalfa 470 [kg/ha]b: coverage [%]

The method of Balázs can be used to characterize the qualitative differences both between different dominant species of grassland and between different grassland types (Table 3). He determined the quality of the species on a scale from −3 to +5 (k). The best quality feed is provided by alfalfa, white clover and red clover. All species that are eaten by the livestock without any harmful consequences for the animal are assigned a + sign and are classified on a scale from +1 to +5. Species that are not eaten by the livestock or their consumption may have harmful consequences are considered as harmful. These species are assigned a − sign and are classified on a scale of −1 to −3. Absolute neutral species between the two groups are assigned a value of 0.

With the knowledge of the average grass height and the total coverage, the annual yield and the animal capacity of the grasslands were estimated. The following data were taken into account: 60 kg/day green weight and 210 day grazing season for cattle, 7 kg/day green weight and 210-day grazing season for sheep and 80 kg/day green weight and 180-day grazing season for horses.

### 4.3. Functional Groups

In order to analyze data easily (Table 4), we classified the surveyed data into functional groups which were as follows: useful grasses, non-useful grasses, legume species, shrub species, invasion species and other species [79,84,85].

### 4.4. Statistical Analysis

To evaluate data clearly, the noticed data were categorized into functional groups which were as follows: useful grasses, non-useful grasses, legume species, shrub species, invasion species and other species. Each area (control, pasture, ridge and areas of Szurdokpüspöki) was described with 6 quadrats botanically representative of the areas, and the data from these quadrats provided the input data for the statistical evaluation per species. The quadrat cover values were summed per quadrat.

The non-parametric statistical method was used to analyze the cover values of species of different groups, as these variables were not normally distributed according to the Shapiro–Wilk test (*p* < 0.05). Accordingly, the non-parametric Kruskal–Wallis test (α = 0.05) was used, and the non-parametric Dunn’s test with Bonferroni correction was used for multiple pairwise comparisons. 

All statistical procedures were performed using the software XL-STAT [86].

## 5. Conclusions

Consequently, based on the presented results, grazing by domestic water buffalo (*Bubalus bubalis*) effected positively on grasslands in both nature conservation and economic aspects. In Central Europe and throughout the Pannonian region in Hungary, on wet and dry grasslands where *Solidago gigantea* was found as a dominant species, grazing by buffalo restored the original plant association, resulting in the re-establishment of the *Molinia* meadows in the wet area. In addition, a typical dry Pannonian grassland association was formed in the ridge-planted area. Therefore, comparing the cover of *Solidago gigantea* in the treated and control areas, we can also consider that water buffalo (*Bubalus bubalis*) is suitable for the eradication of this invasive plant species. On the other hand, in the Szurdokpüpöki area on the Mátra Mountains, buffalo grazing succeeded in restoring the calcareous rocky steppes after the shrub crushing. It prevented properly the encroachment of shrubs into the areas as well as reduced their proportion. Thus, this treatment was adequate for agronomical aspects but at the same time, the habitat also improved in terms of biodiversity.

As a suggestion, we consider that it may be useful to carry on with research to find out more about the effects of grazing by water buffalo (*Bubalus bubalis*) in the current sample areas and also to examine in the future how the grazing by domestic water buffalo (*Bubalus bubalis*) affects the species composition and structure of the grasslands in an azonal area, as there could be significant floristic and physiognomic differences in the vegetation.

It is worth carrying on with the investigation. Considering that the present study has only revealed a little detail about the effects of grazing by buffalo, as it has only been seven years since the observation. It may be interesting to study whether the water buffalo can be used to control other invasive species, as well as to analyze the effects of grazing by buffaloes and other grazing animals in other habitats and to evaluate these econometrically. It would also be advisable to classify according to other aspects what else these animals graze. Furthermore, it may be useful to examine what other effects they have on other animals living there, whether current husbandry technology increases the risk of soil compaction through treading, how manure from buffalo affects wildlife and whether it has a role in the spread of species. All of this and many other aspects can be taken into careful consideration in order to contribute to the establishment of innovative and good grazing practices in the future.

## Figures and Tables

**Figure 1 plants-12-02184-f001:**
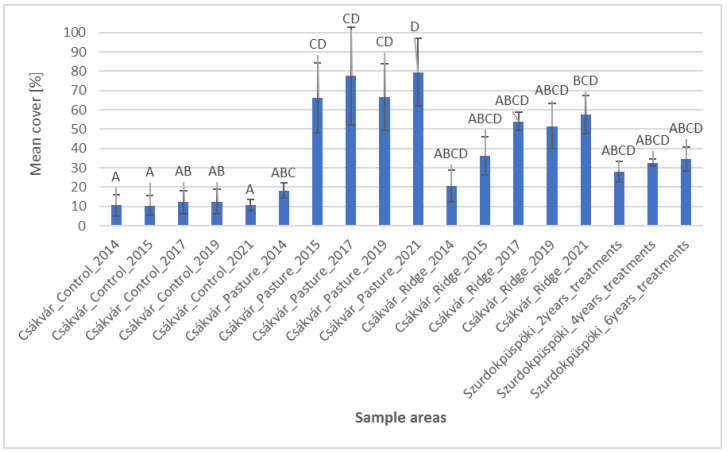
Cover of useful grasses (mean, standard deviation) in the sample areas. Based on statistical evaluation (Kruskal–Wallis test and Dunn’s post hoc test with Bonferoni correction), capital letters indicate homogeneous and heterogeneous groups.

**Figure 2 plants-12-02184-f002:**
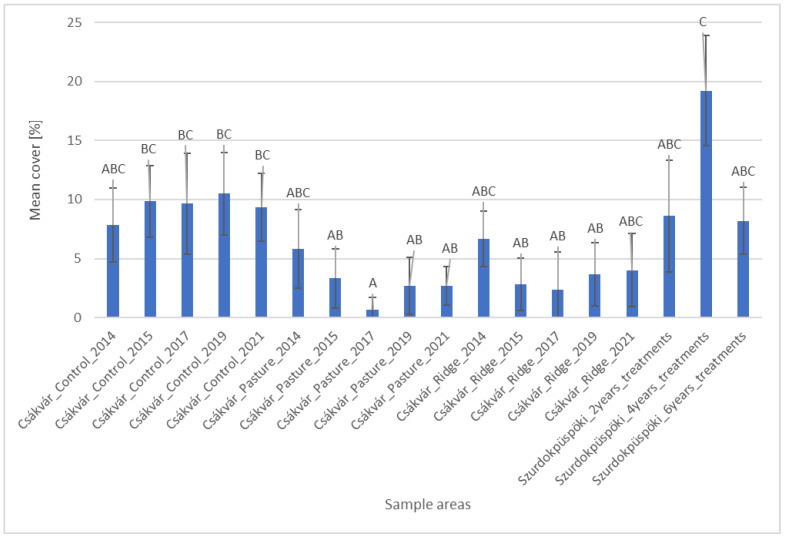
Cover of non-useful grasses (mean, standard deviation) in the sample areas. Based on statistical evaluation (Kruskal–Wallis test and Dunn’s post hoc test with Bonferoni correction), capital letters indicate homogeneous and heterogeneous groups.

**Figure 3 plants-12-02184-f003:**
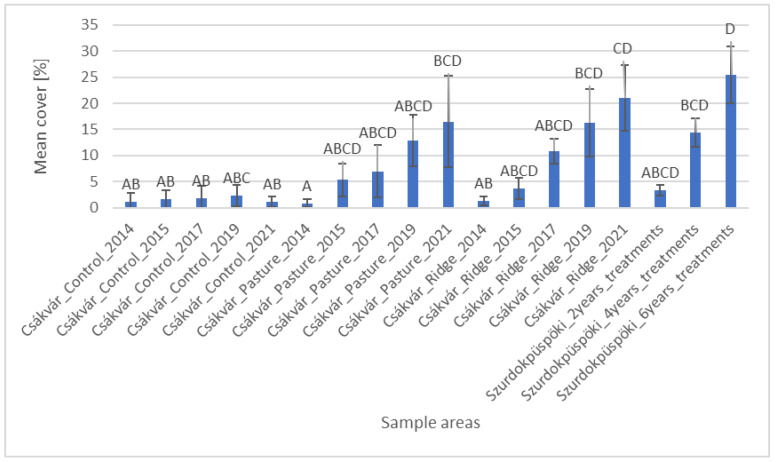
Cover of legume species (mean, standard deviation) in the sample areas. Based on statistical evaluation (Kruskal–Wallis test and Dunn’s post hoc test with Bonferoni correction), capital letters indicate homogeneous and heterogeneous groups.

**Figure 4 plants-12-02184-f004:**
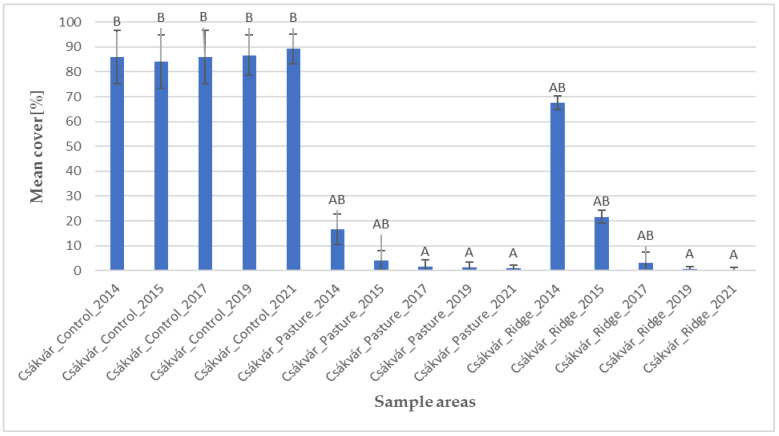
Cover of *Solidago gigantea* (mean, standard deviation) in the sample areas. Based on statistical evaluation (Kruskal–Wallis test and Dunn’s post hoc test with Bonferoni correction), capital letters indicate homogeneous and heterogeneous groups.

**Figure 5 plants-12-02184-f005:**
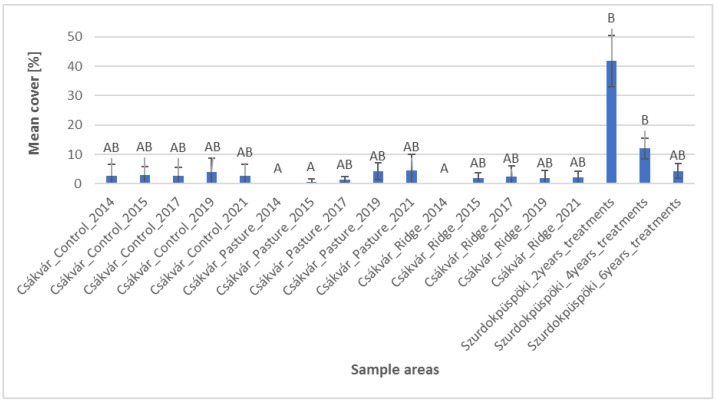
Cover of shrubs (mean, standard deviation) in the sample areas. Based on statistical evaluation (Kruskal–Wallis test and Dunn’s post hoc test with Bonferoni correction), capital letters indicate homogeneous and heterogeneous groups.

**Figure 6 plants-12-02184-f006:**
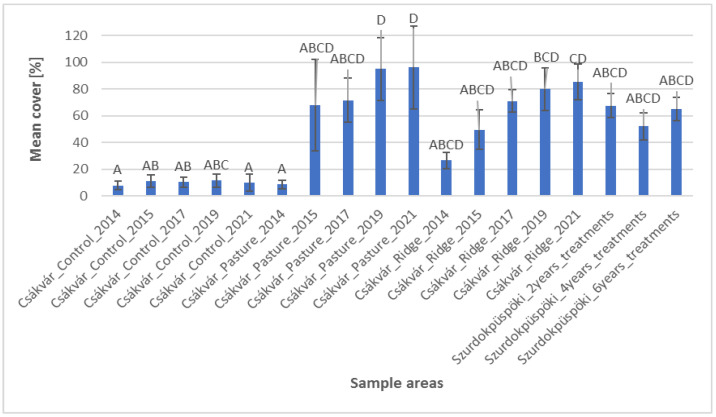
Cover of other species (mean, standard deviation) in the sample areas. Based on statistical evaluation (Kruskal–Wallis test and Dunn’s post hoc test with Bonferoni correction), capital letters indicate homogeneous and heterogeneous groups.

**Figure 7 plants-12-02184-f007:**
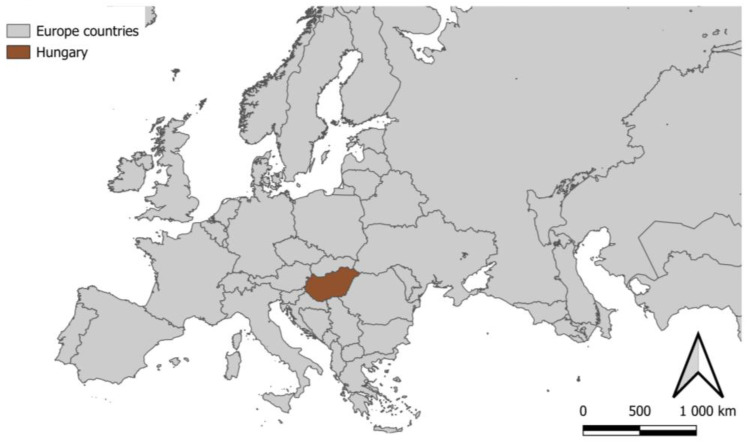
Location of Hungary in the map of Europe.

**Figure 8 plants-12-02184-f008:**
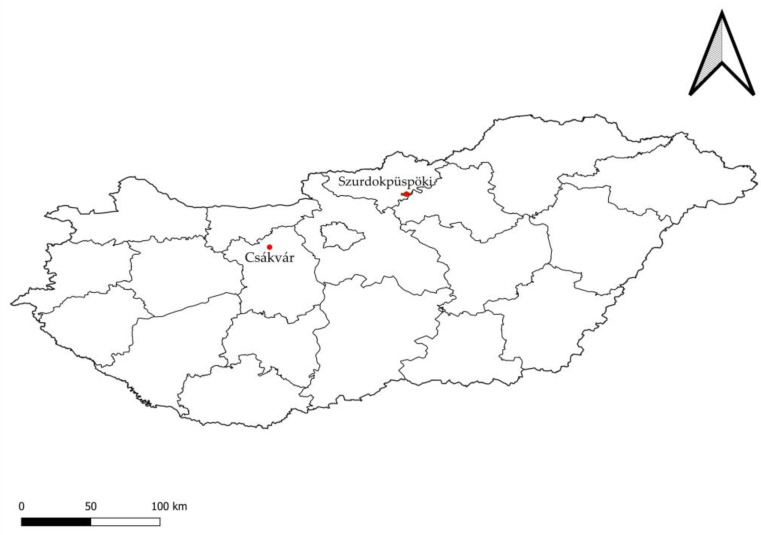
Location of the sample areas in the map of Hungary.

**Table 1 plants-12-02184-t001:** Total cover (%), total yield (t/ha) and feed values of the sample areas in Csákvár (Control area, Pasture, Ridge).

	Control Area				
	2014	2015	2017	2019	2021
Total cover (%) *	116 ±10.13 A **	120 ±12.87 A	122 ±16.01 A	128 ±11.86 A	123 ±10.23 A
Total yield (t/ha)	36.4 ±3.87 A	36.8 ±3.54 A	40.1 ±5.23 A	41.7 ±4.24 A	40.9 ±3.17 A
Feed value	−0.47 ±0.37 A	−0.39 ±0.22 A	−0.30 ±0.29 A	−0.26 ±0.25 A	−0.27 ±0.23 A
	**Pasture**				
	**2014**	**2015**	**2017**	**2019**	**2021**
Total cover (%)	50 ±8.15 A	148 ±47.94 AB	161 36.97 AB	183 ±47.20 B	201 ±58.24 B
Total yield (t/ha)	6.95 ±1.43 A	25.8 ±6.98 AB	26.5 ±6.68 B	27.3 ±7.25 B	15.7 ±4.17 B
Feed value	1.63 ±0.39 A	2.05 ±0.28 AB	2.19 ±0.27 AB	2.16 ±0.24 AB	2.30 ±0.16 B
	**Ridge**				
	**2014**	**2015**	**2017**	**2019**	**2021**
Total cover (%)	123 ±9.87 A	116 ±24.30 A	144 ±9.33 AB	154 ±30.22 AB	170 ±26.56 B
Total yield (t/ha)	28.8 ±1.82 A	19.7 ±3.33 A	19.3 ±1.76 A	19.9 ±3.46 AB	21.2 ±3.07 B
Feed value	−0.62 ±0.21 A	1.87 ±0.17 AB	2.47 ±0.29 B	2.46 ±0.41 B	2.60 ±0.25 B

* The statistical evaluation refers to changes over the years per parameter (Total cover (%), total yield (t/ha) and feed values) and should therefore be interpreted horizontally. ** Capital letters indicate homogeneous and heterogeneous groups.

**Table 2 plants-12-02184-t002:** Total cover (%), total yield (t/ha) and feed values of the sample areas in Szurdokpüspöki (2 years, 4 years, 6 years).

	2 Years	4 Years	6 Years
Total cover (%) *	149 ±13.58 A **	130 ±19.51 A	138 ±19.91 A
Total yield (t/ha)	17 ±1.74 A	14 ±2.03 A	14.3 ±1.95 A
Feed value	1.20 ±0.27 A	2.27 ±0.24 AB	2.64 ±0.30 B

* The statistical evaluation refers to changes over the years per parameter (Total cover (%), total yield (t/ha) and feed values) and should, therefore, be interpreted horizontally. ** Capital letters indicate homogeneous and heterogeneous groups.

**Table 3 plants-12-02184-t003:** Quality classes and their K-values by Balázs.

Classification	Quality of the Feed Value of Grassland	K-Value
Class I.	very good	4<
Class II.	good	3–4
Class III.	medium	2–3
Class IV.	poor	1–2
Class V.	bad	0–1

**Table 4 plants-12-02184-t004:** Classification of species into functional groups [79,84,85].

Economical Useful Grasses					
*Agrostis stolonifera* L.	*Alopecurus pratensis* L.	*Arrhenatherum elatius* (L.) P. Beauv. ex J. Presl & C. Presl	*Brachypodium pinnatum* (L.) P. Beauv.	*Briza media* L.	*Bromus erectus* Huds.
*Bromus inermis* Leyss.	*Dactylis glomerata* L.	*Deschampsia cespitosa* (L.) P. Beauv.	*Elymus repens* (L.) Gould	*Festuca arundinacea* Schreb.	*Festuca ovina* L.
*Festuca pratensis* Huds.	*Festuca pseudovina* Hack.	*Festuca rubra* L.	*Festuca rupicola* Heuff.	*Festuca valesiaca* Schleich. ex Gaudin	*Holcus lanatus* L.
*Molinia caerulea* (L.) Moench	*Phleum pratense* L.	*Poa angustifolia* L.	*Poa humilis* Ehrh. ex Hoffm.	*Sesleria uliginosa* Opiz	
**Economical non-useful grasses**					
*Bothriochloa ischaemum* (L.) Keng	*Bromus arvensis* L.	*Bromus ramosus* Huds.	*Bromus sterilis* L.	*Calamagrostis epigeios* (L.) Roth	*Danthonia decumbens* (L.) DC.
*Cynodon dactylon* (L.) Pers.	*Koeleria cristata* (L.) Pers. em. Borbás ex Domin	*Melica transsilvanica* Schur	*Phragmites australis* (Cav.) Trin. ex Steud.	*Stipa pennata* L.	
**Legume species**					
*Dorycnium germanicum* (Gremli) Rikli	*Dorycnium herbaceum* Vill.	*Genista tinctoria* L.	*Lathyrus nissolia* L.	*Lathyrus pratensis* L.	*Lathyrus tuberosus* L.
*Lotus corniculatus* L.	*Lotus tenuis* Waldst. & Kit. ex Willd.	*Medicago falcata* L.	*Medicago lupulina* L.	*Ononis spinosa* L.	*Tetragonolobus maritimus* (L.) Roth
*Trifolium arvense* L.	*Trifolium montanum* L.	*Trifolium ochroleucon* Huds.	*Trifolium pratense* L.	*Trifolium repens* L.	*Vicia angustifolia* L.
*Vicia cracca* L.	*Vicia tetrasperma* (L.) Schreb.				
**Invasive plants in Hungary**					
*Solidago gigantea* Aiton					
**Shrubs**					
*Crataegus monogyna* Jacq.	*Ligustrum vulgare* L.	*Prunus spinosa* L.	*Pyrus achras* Gaertn.	*Rhamnus cathartica* L.	*Rosa canina* L.
*Rosa gallica* L.	*Rosa rubiginosa* L.	*Rubus caesius* L.			
**Other species**					
*Achillea aspleniifolia* Vent	*Achillea collina* Becker ex Rchb	*Achillea nobilis* L.	*Achillea pannonica* Scheele	*Achillea setacea* Waldst. & Kit.	*Acinos arvensis* (Lam.) Dandy
*Agrimonia eupatoria* L.	*Allium angulosum* L.	*Allium scorodoprasum* L.	*Anchusa officinalis* L.	*Angelica sylvestris* L.	*Arenaria serpyllifolia* L.
*Calystegia sepium* (L.) R. Br.	*Carduus acanthoides* L.	*Carex acutiformis* Ehrh.	*Carex caryophyllea* Latourr.	*Carex distans* L.	*Carex disticha* Huds.
*Carex elata* All.	*Carex flacca* Schreb	*Carex hirta* L.	*Carex melanostachya* M. Bieb. ex Willd.	*Carex panicea* L.	*Carex praecox* Schreb.
*Carex tomentosa* L.	*Centaurea jacea* L.	*Centaurea pannonica* (Heuff.) Hayek	*Cerastium vulgatum* L.	*Cichorium intybus* L.	*Cirsium arvense* (L.) Scop.
*Cirsium canum* (L.) All.	*Colchicum autumnale* L.	*Convolvulus arvensis* L.	*Daucus carota* L.	*Dipsacus laciniatus* L.	*Echium vulgare* L.
*Epilobium parviflorum* Schreb.	*Equisetum arvense* L.	*Erigeron canadensis* L.	*Eryngium campestre* L.	*Eupatorium cannabinum* L.	*Euphorbia cyparissias* L.
*Euphorbia virgata* Waldst. & Kit.	*Filipendula vulgaris* Moench	*Fragaria vesca* L.	*Galium lucidum* All.	*Galium mollugo* L.	*Galium verum* L.
*Gentiana cruciata* L.	*Geranium columbinum* L.	*Hieracium bauhinia* Schult ex Besser	*Hieracium pilosella* L.	*Hieracium sabaudum* L.	*Hypericum perforatum* L.
*Inula britannica* L.	*Inula ensifolia* L.	*Inula salicina* L.	*Iris spuria* L.	*Juncus compressus* Jacq.	*Juncus effusus* L.
*Juncus inflexus* L.	*Lepidium campestre* (L.) W. T. Aiton	*Lepidium draba* L.	*Linaria vulgaris* Mill.	*Lychnis coronaria* (L.) Desr.	*Lycopus europaeus* L.
*Lysimachia nummularia* L.	*Lysimachia vulgaris* L.	*Lythrum salicaria* L.	*Mentha aquatica* L.	*Mentha longifolia* (L.) L.	*Mentha pulegium* L.
*Myosotis arvensis* (L.) Hill	*Odontites rubra* (Baumg.) Opiz	*Anacamptis morio* (L.) R. M. Bateman	Pridgeon & M. W. Chase	*Ornithogalum umbellatum* L.	*Pastinaca sativa* L.
*Pimpinella saxifraga* L.	*Plantago lanceolata* L.	*Plantago major* L.	*Plantago media* L.	*Plantago maritima* L.	*Podospermum canum* C.A.Mey.
*Potentilla anserina* L.	*Potentilla argentea* L.	*Potentilla reptans* L.	*Prunella laciniata* (L.) L.	*Pulicaria dysenterica* (L.) Bernh.	*Ranunculus acris* L.
*Ranunculus polyanthemos* L.	*Ranunculus repens* L.	*Rumex acetosa* L.	*Rumex stenophyllus* Ledeb.	*Sanguisorba officinalis* L.	*Sanguisorba minor* Scop.
*Scirpoides holoschoenus* (L.) Soják	*Senecio aquaticus* Hill	*Serratula tinctoria* L.	*Seseli annuum* L.	*Silene alba* (Mill.) E.H.L.Krause	*Silene bupleuroides* L.
*Silene vulgaris* (Moench) Garcke	*Sonchus arvensis* L.	*Succisa pratensis* Moench	*Taraxacum officinale* F. H. Wigg.	*Thalictrum lucidum* L.	*Thesium linophyllon* L.
*Thymus glabrescens* Willd.	*Tussilago farfara* L.	*Urtica dioica* L.	*Valerianella dentata* (L.) All.	*Verbascum blattaria* L.	*Verbascum phoeniceum* L.
*Veronica arvensis* L.	*Veronica orchidea* Crantz	*Veronica prostrata* L.	*Veronica spicata* L.		

## Data Availability

The data that support the findings of this study are available on request from the corresponding author.

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
