# Peer review of "Examination of the Effects of Domestic Water Buffalo (Bubalus bubalis) Grazing on Wetland and Dry Grassland Habitats"

_plants, 2023, doi:10.3390/plants12112184_

Round 1

Reviewer 1 Report

Comments to the paper manuscript, entitled „Examination of the Effects of Domestic Water Buffalo (Bubalus bubalis) Grazing on Wetland and Dry Grassland Habitats”.

 The objective of the present study was to evaluate the impact of water buffalo grazing on biodiversity improvement of three types of valuable grassland vegetation. Extensive grazing is well-known measurement for maintenance of biodiversity of grassland habitats. However, new findings on the effect of grazing of water buffalo on botanical structure of Wetland and Dry grassland habitats Pannonian grasslands and suppression of Solidago gigantea are important.

 I have some comments to the paper.

 Abstract

Lines 34 -36: Authors state that in the Zámolyi Basin, Sesleria uliginosa has started to dominate in the grassland. But there are not any botanical surveys in the paper to support this finding.

Material and methods

-   - Authors compare three grassland types. But the length of grazing is stated only for the grasslands in Szurdokpüspök in Mátra Mountain. It would be good to specify how many years the grasslands and ridge in Csákvár were grazed.

-    - To evaluate the effect of grazing it would be also good to know the stocking rate in each grassland type.

-     -  The information on grazing period.

Author Response

Dear reviewer,

Thank you very much for your comments. We completed the manuscript with the necessary information based on your review.

  • We completed the results in order to harmonise them with the abstract data.
  • Grazing in Csákvár has been carried out since 2013. It is corrected in the manuscript, too.
  • The information on grazing period is completed in the manuscript.

Reviewer 2 Report

Manuscript review

"Examination of the Effects of Domestic Water Buffalo (Bubalus 2 bubalis) Grazing on Wetland and Dry Grassland Habitats" Plants 2341460

For authors

The study submitted for review describes in an interesting way the impact of water buffaloes on vegetation in different habitat types leading to elimination of the invasive species, improvement of degraded grassland yields and forage quality. The study was carried out in Europe, in 4 different areas of Hungary: in the Matra Mountains (grassland areas where grazing had been appled for 2, 4, 6 years); in the Zamolyi Basin dominated by the invasive plant Solidago gigantea and in the Pannonian dry grassland area grazed by water Buffalo (Bubalus bubalis).

The authors studied the change of cover of plant species, their feed values and biomass of the grassland in control areas and in areas grazed by water buffalo (Bubalus bubalis).

An increase in the coverage (%) of the observed degraded grassland areas with valuable, economically important grasses was shown. The emergence of papilionaceous plants, low soil cover by shrubs and an increasing proportion of grasses in the meadows over time were observed. The invasive plant Solidago giganta became almost completely extinct under the influence of water buffalo grazing, and pasture grasses developed on the site it left behind, with Selsaria uliginosa being the dominant species. 

In the reviewed manuscript, the literature review is quite extensive, interesting and well conducted. This chapter cites 74 pieces of literature related to the research topic. However, what is missing from this chapter is a research hypothesis on the impact of water buffalo grazing on vegetation during the conversion of areas controlled by invasive plants, shrubs and trees into semi-natural grasslands. The research hypothesis should be included in the text before the research objective which is well formulated in this study.

Chapter Results. Consider including the names of all species belonging to functional groups in the tables (lines 122-176). Include explanations for the letters A, AB, B, CD, etc. in all figures. It is necessary to explain the significance of differences between the compared results and the meaning of the error bars in Figures 1-6. Complete the data on the significance of the results concerning the degree of cover and plant biomass in Tables 1 and 2.

Change the description of the data discussed in lines 262-268, because in my opinion it is not possible to discuss test results whose value is not known, neither shown in the figure nor in the table. If possible, I suggest including the yield values of all species in the table, which can then be discussed in detail.

Methodology. The methodology in the coenological surveys section is understandable. The question that comes to my mind is "couldn't the authors, when taking the phytosociological images, have cut a sample of the vegetation to determine the actual biomass" (weighed biomass). I understand that it is difficult to sample such vegetation when it is shrubs and trees, however, would it not have been possible to try to assess biomass yield in this way. The authors assessed the biomass of plants produced in meadows according to the Balazs methodology (literature items 93 and 94). I tried to read the details of this methodology, but this was impossible because none of the papers cited were available electronically. Perhaps there are other, less complicated and newer methods for evaluating meadow biomass in such studies that could be used when discussing the results in this experiment.

Statistical calculations. The authors used a little-known computer programme to compare the averages and show significant differences between them. Most often in scientific papers, the program Stalistica is used to prove significant variation between the obtained test results. I wonder why the authors did not carry out a statistical analysis for biomass yield and feed value? This analysis and the inclusion of its results in Tables 1 and 2 will give a reliable picture of the changes over the years in the yield and feed value of grasslands under the influence of water buffalo grazing.

Conclusions. Do not generalise the conclusions obtained but specify under which conditions water buffaloes reduced the growth of the invasive species Solidago gigantea and shrubs and led to the restoration of semi-natural grasslands on grazed land. That is, specific data related to the location of the study (country, region, habitat conditions) should be provided.

The authors' suggestion of further research into the use of water buffalo for the rehabilitation of degraded grasslands and the control of invasive vegetation is a good one. I know from my own practice the positive effects of grazing the Scottish Highland cattle breed in destroying shrubs and some trees and unnecessary vegetation in year-round pastures in Poland.

Literature. The authors presented an extensive literature review of 96 items.  All publications included in the list of publications were cited by the authors in the text. It should be emphasised that 61 items of literature cited in this manuscript are papers published after 2010, i.e. the authors, while preparing this manuscript, used the latest literature on restoring degraded grasslands to semi-natural grasslands. I suggest that the authors refrain from citing some of the older or not very relevant to the research described in the literature list.

The reviewer has included some of the comments in the pdf manuscript.

In conclusion, I believe that the paper presented for review is interesting, providing practical guidance on how the restoration of grasslands degraded to semi-natural pasture should be carried out using water buffalo grazing. The work needs some additions and minor corrections and additional statistical calculations (data from Tables 1 and 2). With some corrections and additions to the data, the paper is suitable for publication in a prestigious journal such as Plants. The reviewer's comments aim to improve the quality of the manuscript and to better understand the research results described.

Thank you for your attention

Reviewer

Author Response

Dear reviewer,

Thank you very much for your comments.

We have completed the manuscript with hypotheses, which is really necessary in the manuscript. A hypothesis based on the reviewed literature is that as buffalo have a better digestive capacity than cattle, they may be able to graze on giant goldenrod with a high saponin content and woody plants such as shrubs. It would be necessary because Solidago gigantea suppresses plants that are useful for turf management, too.

All species of the functional groups are already presented in a table and the statistical figures are completed with an explanatory legend. We have also added the necessary results of the statistical calculations to the first and second tables.

Lines 262-268: It is corrected with the addition of supplementary materials.

Metholodgy: The reason why we did not collect cut samples for the biomass analysis was because we only did coenological surveys at the beginning of the study, the idea of biomass analysis emerged later, and therefore we did not collect biomass samples otherwise we would not have been able to compare the biomass data correctly. The method of Balázs is similar to the method of Klapp et al., but the method of Balázs is extended with feed quality values, so its use is essential. To understand this method, we added a description to the manuscript.

Conclusion. The conclusions have been corrected in the manuscript.

Literature. It is corrected.

Reviewer 3 Report

The topic would certainly have the potential to make an interesting contribution to the special issue. However, the present manuscript must be rejected in this form, as it does not meet the minimum requirements of the journal. ‘Plants’ is a highly ranked journal, so the large number of authors should simply have made more of an effort.

The deficits concern the lack of precision in the use of language, which reveal not only linguistic but also intellectual weaknesses. For example, the reader does not learn what the authors mean by improving the plant population at all: are they concerned with a higher fodder value (agronomic improvement) or are they concerned with plant sociological enhancement (biodiversity and species protection). Distinguishing between the two is essential for understanding the entire study.

The introduction describes both the landscape conservation potential of water buffalo grazing and the danger of Solidago invasion, but the connection was not elaborated. Are water buffaloes expected to graze this poisonous plant? If so, a veterinary study should have been carried out to monitor the health of the animals. Or is the invasive species to be pushed back by kicking and encouraging potential competitors? Then one would have expected hypotheses on the mechanisms of this process. Furthermore, the lack of studies on the feeding ecology of the water buffalo in Europe is deplored, but substantial sources (author search for "Sweers" in the relevant databases) have not been researched.

The classification of species into the individual functional groups belongs in the methods section and not in the results. It would be necessary to make the criteria of the classifications transparent. Other methods are also not comprehensible and do not meet international standards (e.g., biomass estimation and estimation of feed value without state-of-the art laboratory feed analysis). Ultimately, the data situation is limited to the coverage rates, whose comparisons across locations and years are only really reliable under ceteris paribus conditions. The illustrations were also made rather uncharitably; the possibilities of colour choices for locations and duration of use were not used, and the labels were confusing and redundant.

The grazing preference of woody plants should not be discussed independently of grazing pressure (stocking density) as a main covariate.

Author Response

Dear reviewer,

Thank you very much for your comments. We have made several corrections to the manuscript.

The primary objective was basically agronomic, in terms of turf management, habitat improvement was important, and in the case of the grasslands studied, especially in the Solidago populations, it was very related to species diversity, because Solidago gigantea supresses species of importance for turf management. In addition to the agronomic improvement, the species diversity of the area was also improved. 

To describe the connection between Solidago and the buffalo, we have added a hypothesis to the introduction.

There is a meat packing plant on the areas, so the animals are monitored regularly for health reasons, and therefore it was not necessary to carry out any specific veterinary studies.

All species of the functional groups are already presented in a table in the correct section and the statistical figures are completed with an explanatory legend. We have also added the necessary results of the statistical calculations to the first and second tables.

Metholodgy: The method of Balázs is similar to the method of Klapp et al., but the method of Balázs is extended with feed quality values, so its use is not essential. To understand this method, we added a description to the manuscript. We have made the necessary corrections to make the manuscript internationally compatible.

The use of ceteris paribus in our present study is difficult to implement.

Stocking density is already completed in the manuscript.